

# A global high-resolution data set of ice sheet topography, cavity geometry and ocean bathymetry

Janin Schaffer[1], Ralph Timmermann[1], Jan Erik Arndt[1], Steen Savstrup Kristensen[2], Christoph Mayer[3], Mathieu Morlighem[4], and Daniel Steinhage[1]

[1]Alfred Wegener Institute, Helmholtz Centre for Polar and Marine Research, Bremerhaven, Germany
[2]DTU Technical University of Denmark, 2800 Lyngby, Denmark
[3]Bavarian Academy of Sciences and Humanities, Commission for Geodesy and Glaciology, Munich, Germany
[4]University of California, Irvine, Department of Earth System Science, Croul Hall, Irvine, California 92697-3100, USA

*Correspondence to:* J. Schaffer (Janin.Schaffer@awi.de) and R. Timmermann (Ralph.Timmermann@awi.de)

**Abstract.** The ocean plays an important role in modulating the mass balance of the polar ice sheets by interacting with the ice shelves in Antarctica and with the marine-terminating outlet glaciers in Greenland. Given that the flux of warm water onto the continental shelf and into the sub-ice cavities is steered by complex bathymetry, a detailed topography data set is an essential ingredient for models that address ice-ocean interaction. We followed the spirit of the global RTopo-1 data set and compiled consistent maps of global ocean bathymetry, upper and lower ice surface topographies and global surface height on a spherical grid with now 30-arc seconds resolution. We used the General Bathymetric Chart of the Oceans (GEBCO_2014) as the backbone and added the International Bathymetric Chart of the Arctic Ocean version 3 (IBCAOv3) and the International Bathymetric Chart of the Southern Ocean (IBCSO) version 1. While RTopo-1 primarily aimed at a good and consistent representation of the Antarctic ice sheet, ice shelves and sub-ice cavities, RTopo-2 now also contains ice topographies of the Greenland ice sheet and outlet glaciers. In particular, we aimed at a good representation of the fjord and shelf bathymetry surrounding the Greenland continent. We corrected data from earlier gridded products in the areas of Petermann Glacier, Hagen Bræ and Sermilik Fjord assuming that sub-ice and fjord bathymetries roughly follow plausible Last Glacial Maximum ice flow patterns. For the continental shelf off northeast Greenland and the floating ice tongue of Nioghalvfjerdsfjorden Glacier at about 79°N, we incorporated a high-resolution digital bathymetry model considering original multibeam survey data for the region. Radar data for surface topographies of the floating ice tongues of Nioghalvfjerdsfjorden Glacier and Zachariæ Isstrøm have been obtained from the data centers of Technical University of Denmark (DTU), Operation Icebridge (NASA/NSF) and Alfred Wegener Institute (AWI). For the Antarctic ice sheet/ice shelves, RTopo-2 largely relies on the Bedmap-2 product but applies corrections for the geometry of Getz, Abbot and Fimbul ice shelf cavities. The data set is available in full and in regional subsets in NetCDF format from the PANGAEA database at doi:10.1594/PANGAEA.856844.

## 1 Introduction

Mass loss from the Greenland Ice Sheet presently accounts for about 10% of the observed global mean sea-level rise (Church et al., 2013). The ocean plays an important role in modulating the flow of ice by delivering heat to the marine-terminating outlet



glaciers around Greenland (e.g., Seale et al., 2011; Straneo et al., 2012). The warming and accumulation of Atlantic Water in the subpolar North Atlantic has been suggested to be the driver of the glaciers' retreat around the coast of Greenland (e.g., Straneo and Heimbach, 2013). The complex bathymetry in this region is thought to steer the flux of warm water of Atlantic origin from the open ocean onto the continental shelf towards the calving fronts of outlet glaciers and into the cavity below

floating ice tongues. This is particularly true for the northeast Greenland continental shelf, where a system of troughs provides a flux of warm water towards the floating ice tongues of Nioghalvfjerdsfjorden Glacier (also referred to as 79°North Glacier) and Zachariæ Isstrøm (Arndt et al. (2015), Wilson and Straneo (2015)). Recently, these glaciers were observed to retreat and melt rapidly (Mouginot et al., 2015). In such regions detailed bathymetry data and consistent data sets of ice topographies are essential ingredients for studying the interaction between the ocean and the cryosphere.

The Refined Topography data set (RTopo-1, Timmermann et al. (2010)) provides consistent maps of the global ocean bathymetry and the upper and lower ice surface topographies of the Antarctic ice sheet and shelves. Horizontal resolution of these maps is 1-arc minute. Based on RTopo-1, ocean general circulation models have successfully been used to simulate, e.g., Southern Ocean warming and increased ice shelf basal melting around Antarctica (Timmermann and Hellmer, 2013; Kusahara and Hasumi, 2013), the flow of warm deep water onto the Amundsen Sea continental shelf ( Assmann et al. (2013), Nakayama

et al. (2014a)), and pathways of basal meltwater from Antarctic ice shelves (Kusahara and Hasumi (2014), Nakayama et al. (2014b)). Parts of RTopo-1 were used to compile improved maps of bedrock and ice topographies for Antarctica in Bedmap2 (Fretwell et al., 2013). The Greenland Ice Sheet, however, has remained a blank area in RTopo-1.

The aim of this paper is to present the newly compiled global topography data set RTopo-2, which provides a detailed bathymetry for the continental shelf around Greenland and contains ice and bedrock surface topographies for Greenland and

Antarctica as part of a global, self-consistent data set with a horizontal resolution of 30-arc seconds. In the following sections, we introduce the data used, the processing applied to each data set and the strategies followed for merging the data sets in a self-consistent way. We demonstrate the improvements achieved in RTopo-2 compared to previous products and discuss the most relevant caveats.

## 2 Data sets and processing

### 2.1 Overview of RTopo-2 maps

We followed the spirit of RTopo-1 and compiled global fields for

1. bedrock topography (ocean bathymetry; surface topography of continents; bedrock topography under grounded or floating ice)

2. surface elevation (upper ice surface height for Antarctic and Greenland ice sheets/ice shelves; bedrock elevation for
ice-free continent; zero for ocean)

3. ice base topography for the Antarctic and Greenland ice sheets/ice shelves (ice draft for ice shelves and floating glaciers; zero in absence of ice)



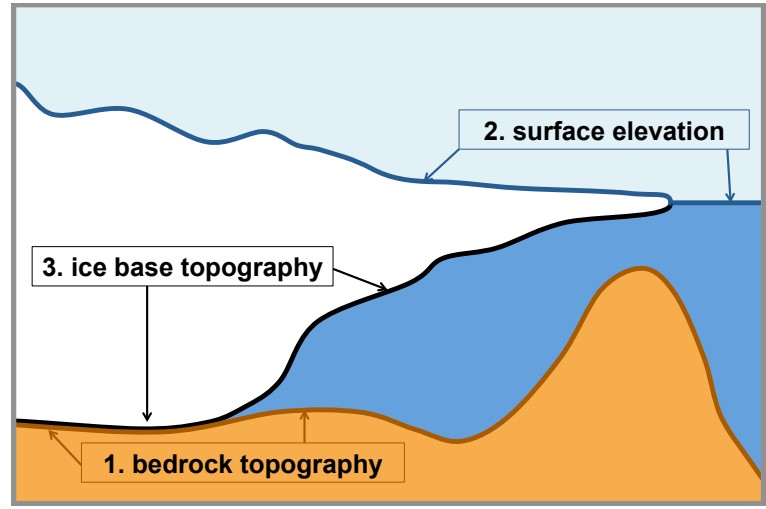

**Figure 1.** Sketch of a 2D vertical section along a floating glacier tongue/ice shelf with grounded and floating ice (white), sub-ice bedrock/ocean seafloor (brown), water in a subglacial cavity and the open ocean (blue). Lines indicate the bedrock topography (brown), the surface elevation (dark blue), and the ice base topography (black).

4. a surface type mask that indicates open ocean, grounded ice (ice sheets), floating ice (ice shelves/floating glaciers), and bare land surface

5. positions of coastlines and ice shelf/floating glacier front lines.

The bedrock topography is identical to the surface elevation for ice-free land surface, and identical to the ice base topography
for grounded ice (Fig. 1). Ice not connected to the Greenland or Antarctic ice sheet is not covered in our data set. Glaciers on subantarctic and Greenland islands are thus labeled as bare land surface with the surface elevation preserved. Lakes were deleted and handled as bare land surface; subglacial lakes on the Antarctic continent, however, have been preserved. In contrast to RTopo-1, we now provide all maps with a horizontal resolution of 30-arc seconds.

## 2.2 Data sources and merging strategies

RTopo-2 has been compiled by combining various gridded data sets (Table 1) with different resolutions, projections, and coverages (Fig. 2) into global maps. Interpolation of the source data sets from their different projections to our geographic grid was done by triangulation; a careful smoothing was applied to avoid artefacts.

### 2.2.1 World Ocean bathymetry

As the nucleus of RTopo-2 we use an updated version of the General Bathymetric Chart of the Oceans (GEBCO) 30 arc-second
data set (Becker et al., 2009), namely the GEBCO_2014 (20150318) Grid that was released in March 2015 (Weatherall et al., 2015). The global grid of seafloor elevations is based on quality-controlled ship depth soundings. In between soundings the in-



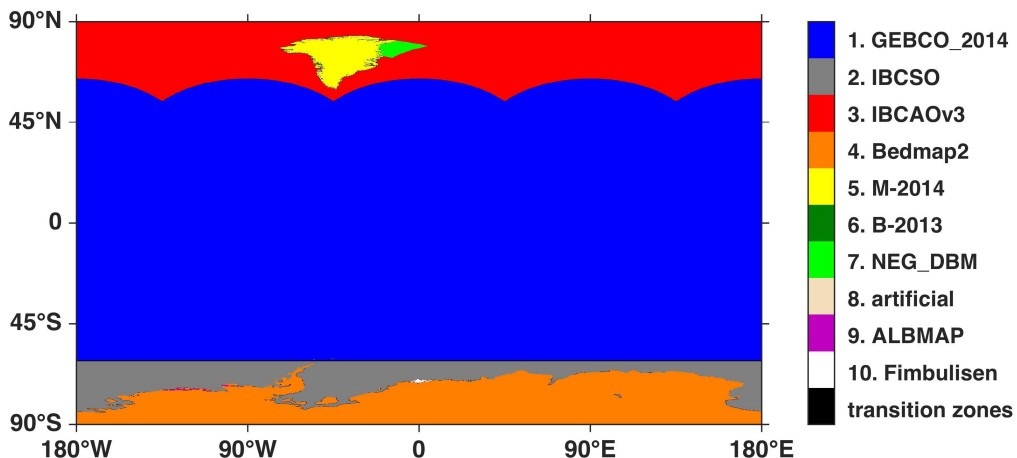

**Figure 2.** Data sources for the ocean bathymetry in RTopo-2. Black areas denote transition zones between the data sets (source flag 20). Further explanations are given in Table 1.

terpolation was guided by satellite-derived gravity data (Sandwell and Smith, 2009). In some areas, GEBCO_2014 furthermore uses the local expertise of regional undersea mapping projects (see next chapter).

### 2.2.2  Southern and Arctic Ocean bathymetries

GEBCO_2014 includes the International Bathymetric Chart of the Southern Ocean (IBCSO) Version 1.0 (Arndt et al., 2013)
south of 60°S and the latest version of the International Bathymetric Chart of the Arctic Ocean (IBCAOv3) (Jakobsson et al., 2012) north of 64°N. Ice sheets, floating ice shelves and glaciers in GEBCO_2014 are represented by their surface elevation, which in case of the Antarctic ice sheet has been adopted from the IBCSO 'ice surface' grid. Given that we aim at a continuous representation of the sub-ice cavities as part of the global ocean, we replaced the GEBCO data by the IBCSO 'bedrock topography' grid south of 61.5°(Fig. 2). Towards the Arctic Ocean, we re-combined GEBCO_2014 with IBCAOv3 (Fig. 2) in
order to keep the high-resolution information from multibeam surveys off the southern tip of Greenland (south of 64°N) which are included in IBCAOv3 (Jakobsson et al., 2012) but have not been adopted in GEBCO. Both digital bathymetry products, IBCAOv3 and IBCSO, have a horizontal resolution of 500 m x 500 m and were constructed from a combination of all multibeam, dense single beam and land surface height data available for these regions. A smooth blending of 50 km/20 km is applied along the transition lines between GEBCO and IBCSO/IBCAOv3.

**2.2.3  Ocean bathymetry in Greenland fjord and continental shelf regions**

The bathymetry of the continental shelf along the Greenland coast is crucial to ice-ocean studies in this region and thus there is a rising interest in a good representation of these areas. Nevertheless, away from the commonly used ship routes and especially



**Table 1.** Data sources for individual regions merged in RTopo-2. The index numbers correspond to the source flags in Fig. 2.

| | Region | Data obtained from |
|---|---|---|
| 1. | World Ocean bathymetry | GEBCO_2014 (Weatherall et al., 2015) |
| 2. | Southern Ocean bathymetry | IBCSO (Arndt et al., 2013) |
| 3. | Arctic Ocean bathymetry | IBCAOv3 (Jakobsson et al., 2012) |
| 4. | Antarctic ice sheet/shelves surface height and thickness and bedrock topography | Bedmap2 (Fretwell et al., 2013) |
| 5. | Greenland ice sheet/glaciers surface height and thickness and bedrock topography | Morlighem et al. (2014) (M-2014) |
| 6. | Fjord and shelf bathymetry close to the Greenland coast | Bamber et al. (2013) (B-2013) |
| 7. | Bathymetry on Northeast Greenland continental shelf | Arndt et al. (2015) (NEG_DBM) |
| 8. | Bathymetry in several narrow Greenland fjords and on parts of the Greenland continental shelf | artificial, see Merging strategy and Data corrections in Section 2.2.3 for details |
| 9. | Bathymetry for Getz and western Abbot Ice Shelf cavities | ALBMAP (Le Brocq et al., 2010) |
| 10. | Bathymetry for Fimbulisen cavity | Nøst (2004), Smedsrud et al. (2006) |
| 11. | Ice thickness for Nioghalvfjerdsfjorden Glacier and Zachariæ Isstrøm | DTU (Seroussi et al., 2011) Operation Icebridge (Allen et al., 2010, updated 2015) Alfred Wegener Institute (AWI) Mayer et al. (2000) |
| 12. | Contour of iceberg A-23A in Weddell Sea | Paul et al. (2015) |

in ice-covered areas, the depth of the sea floor is only weakly constrained. Data coverage maps of IBCAOv3 show that many shelf and fjord areas around the coast of Greenland are not covered by soundings (Jakobsson et al., 2012). To achieve a more detailed representation of Greenland continental shelf bathymetry, we included additional data sources (Table 1).

**B-2013: Bedrock topography from Bamber et al. (2013)**

Based on surface elevation maps from the Greenland Iceland Mapping Project (GIMP, Howat et al. (2014)) and ice thickness data from multiple airborne surveys between 1970 to 2012, Bamber et al. (2013) compiled a data set of ice thickness and bedrock elevation on and around Greenland (B-2013). Using the ocean bathymetry from IBCAOv3 and with plausible

10 assumptions for historic glacier ice flow pathways, the bottom topography was modified in several places to achieve a better representation of the fjord structures and of the troughs that connect the fjords to the continental shelf break. While this was



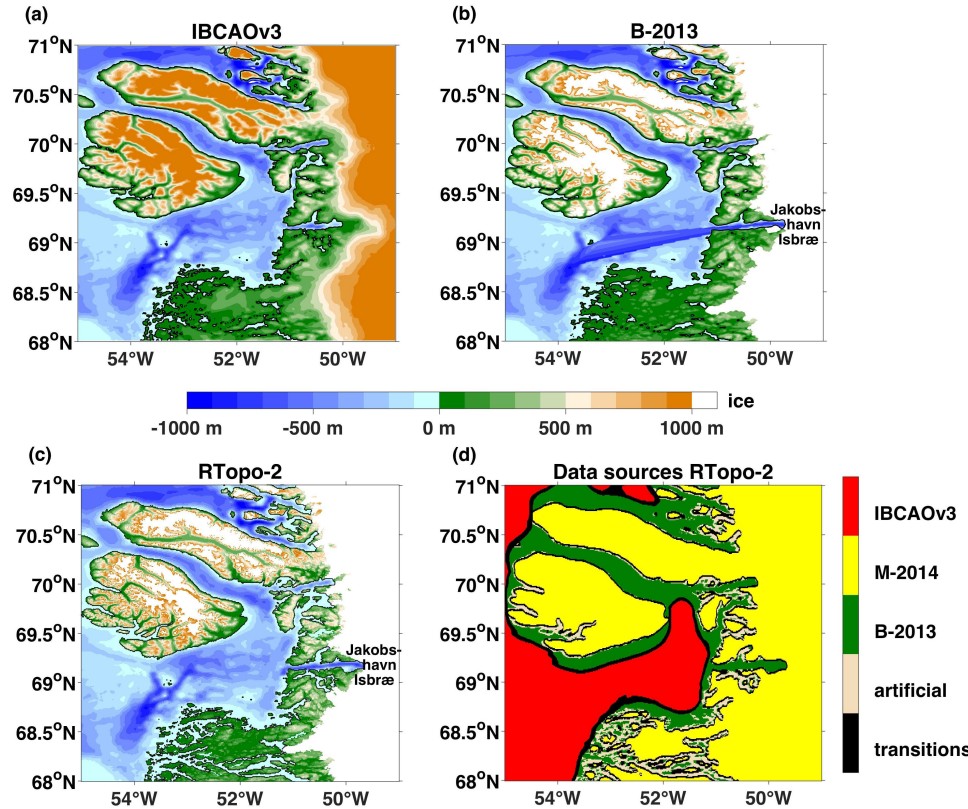

**Figure 3.** Coastal and shelf region in the west of Greenland including Jakobshavn Isbræ. Maps show the ocean bathymetry/surface elevation in IBCAOv3 (a), ocean bathymetry/bedrock elevation in B-2013 (b), ocean bathymetry/bedrock elevation in RTopo-2 (c) and the data sources for RTopo-2 (d). The color scale is identical for all bathymetry maps. White shading indicates grounded ice and floating ice tongues; black lines mark the coastline. The color flags of the different data sources are identical to Fig. 2.

clearly an important step towards a better representation of the relevant processes in models, the B-2013 data set has three major weaknesses. First, lines of very steep gradients still indicate an unrealistic bedrock topography where the fjord's bathymetry was interpolated/extrapolated across the shelf (e.g., Fig. 3b). Second, many of the smaller fjords are not resolved due to the 1 km-resolution of the data set. Lastly, the different maps in this data set are not fully consistent with each other; combining surface elevation and ice thickness maps yields an ice bottom topography that even for grounded ice is not identical to the bedrock topography grid provided.

**M-2014: Bedrock topography from Morlighem et al. (2014)**
In addition to the airborne ice thickness survey data and the surface elevation obtained from GIMP (Howat et al., 2014), the Morlighem et al. (2014) (M-2014) data set also considers satellite-derived ice motion data and applies a mass conservation



scheme to derive an ice thickness distribution that is consistent with the observed flow lines. Like in B-2013, the bedrock elevation was calculated by subtracting the ice thickness from the surface elevation. While the resulting topography data set in M-2014 does not contain any information for ocean areas, it still provides very useful guidance for fjord structure and topography. The distribution of grounded/floating ice and bare land in M-2014 follows the GIMP coastline and thus represents even

the smaller fjords with a lot of detail. Morlighem et al. (2014, suppl.) showed that many ice-covered and open-ocean fjords are not resolved in the Bamber et al. (2013) data set. We used the land/sea/ice mask from M-2014 as the most important criterion for merging the different bathymetry data sets and applying corrections to the data.

**Merging strategy**

To benefit from the best parts of each data set, we used

- the bedrock elevation from M-2014 for all locations with grounded ice,

- the bed elevation from B-2013 within the fjords and in a narrow band of about 25 km width along the Greenland coast,

- the bathymetry from IBCAOv3 further away from the coast, with transition zones of 10 km width.

Consequently the large areas of continental shelf around Greenland are mainly determined by IBCAOv3 data while the fjord topographies are given by the B-2013 bedrock (e.g., Fig. 3d). In regions where the GIMP coastline demands ocean but B-2013 gives land values, we prescribed small patches with negative topography values (source flag 8, e.g., Fig. 3d). The depth of these artificial points was chosen to be 10 m for grid points right next to land and 100 m for grid points along the center of the fjords. These small patches of artificial values were smoothed with their surroundings to obtain plausible shapes of bedrock

topography (e.g., Fig 3c).

**Data corrections**

Three sectors turned out to be particularly difficult to handle:

1. the region around Petermann and Ryder Glaciers (North Greenland)

2. the North Greenland fjords system off Hagen Bræ and Marie Sophie and Academy Glaciers, and

3. the Sermilik Fjord in front of Helheim, Fenris and Midgaard Glaciers, and Køge Bugt (Southeast Greenland).

Observations at the front of Petermann Glacier's floating ice tongue imply that the subglacial fjord is about 900 m deep (Johnson et al., 2011) as opposed to about 400 m in B-2013. The observations cover only a small area at the glacier front; no information for the subglacial bathymetry towards the grounding line of the floating ice tongue is available. We deepened and

smoothed the subglacial fjord to a maximum depth of about 500 m, which is more likely to under- than to overestimate the true depth. Further to the east, the B-2013 representation of the continental shelf area in front of Ryder Glacier features a very steep gradient and a deep trough close to the coast, which appears unrealistic. We replaced some of the interpolated deep and



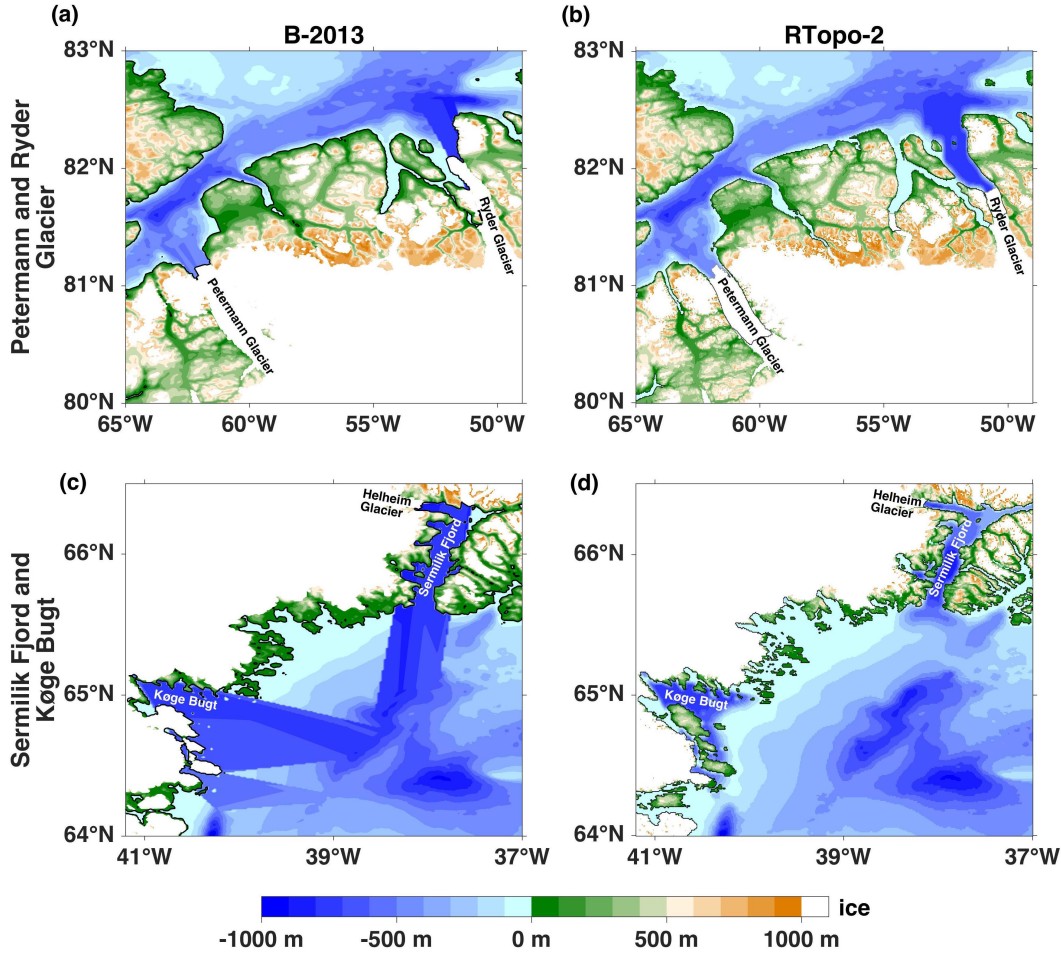

**Figure 4.** Maps of the ocean bathymetry/bedrock elevation in B-2013 (a, c) and RTopo-2 (b, d). The upper panel (a, b) shows the coastal and shelf region in the northern sector of Greenland including the fjord system in front of Petermann Glacier and Ryder Glacier. The lower panel (c, d) gives the coastal and shelf region in the southeastern sector of Greenland including the Sermilik Fjord and Køge Bugt. White shading indicates grounded ice and floating ice tongues; black lines mark the coastline.

adjacent shallow parts with a smooth deep fjord/shelf bathymetry (Figs. 4a and 4b). We think that this gives a more plausible representation compared to B-2013 although it needs to be kept in mind that there is no observational evidence for either case.

For the second region, the fjord system in front of the Hagen Bræ, Marie Sophie and Academy Glaciers, we defined small patches of artificial topography to achieve a smooth transition between the subglacial bedrock and the fjord bathymetry at the
5 glacier fronts. We connected the under-ice bathymetry with the ocean bathymetry following plausible Last Glacial Maximum (LGM) ice flow patterns. The LGM ice sheet margin was approximately located at the continental shelf break in this region (Funder et al., 2011). In addition, we smoothed the ocean bathymetry for the fjord system to remove steep artificial gradients arising from B-2013.





For the Sermilik fjord off Helheim Glacier and for Køge Bugt, the bathymetry data from B-2013 show very deep troughs and steep gradients on the continental shelf (Fig. 4c). We inserted artificial values in several locations in the fjords and smoothed over the relevant part of the grid. The result (Fig. 4d) is to a large extent consistent with the observations of Andresen et al. (2012) in Sermilik Fjord.

All regions with inserted artificial values were marked with the data source flag 8.

### 2.2.4 Bathymetry of the Northeast Greenland continental shelf

Bottom topography on the continental shelf northeast of Greenland is poorly resolved and contains a number of artifacts in IBCAOv3. Reprocessing and combining multi- and single-beam echosounding data from more than two decades resulted in a significantly improved digital bathymetry model (NEG_DBM) (Arndt et al., 2015). In addition to the echosounding data,
maximum depths from CTD profiles were included in areas with no other available information.

We included the NEG_DBM bedrock elevation in the continental shelf area between the Greenland coast in the west and the continental shelf break (600 m depth contour) in the east, from 75°N to about 80.5°N (Fig. 2). The coastline of the main land remains based on M-2014/GIMP, while the topography and coastline of the islands in this area were adopted from the NEG_DBM.

Within the NEG_DBM the sub-ice bathymetry of Nioghalvfjerdsfjorden Glacier is interpolated based on seismic observations from Mayer et al. (2000). We expect the sub-ice bathymetry to roughly follow plausible LGM ice flow stream lines (Evans et al., 2009) which we inferred from the seismic data points. We adjusted the interpolated bathymetry accordingly to achieve a more realistic representation of the sub-ice cavity geometry.

### 2.3 Ice and bedrock topographies

### 2.3.1 Greenland ice and bedrock topographies

As already mentioned in Section 2.2.3, Morlighem et al. (2014) combined (1) the surface elevation obtained within GIMP, (2) data from airborne ice thickness surveys, and (3) satellite-derived ice motion data to provide high-resolution maps of ice thickness and bedrock topography for the Greenland ice sheet. Using mass conservation as a constraint in the optimization, an ice thickness distribution that is consistent with the observed flow lines was proposed. Given that ice bottom topography
was calculated by subtracting the ice thickness from the surface elevation, this also affects the representation of bedrock for grounded ice areas. The M-2014 maps extend to the ice front in case of grounded ice and to the coastline for bare land. For floating ice, the bedrock elevation extends only to the grounding lines while ice thickness and surface elevation data cover the full area to the ice front. With a horizontal resolution of 150 m, the data set resolves many more fine-scale structures than the 1-km B-2013 product (Morlighem et al., 2014, suppl.).

We use M-2014 as the backbone representation of the Greenland Ice Sheet geometry and as the basis for the ice/land/sea mask within the perimeter of the Greenland continent. These are based on ocean and ice masks from GIMP, while the ice shelves were added by using InSAR mapping (differential satellite radar interferometry) following Rignot et al. (2011).





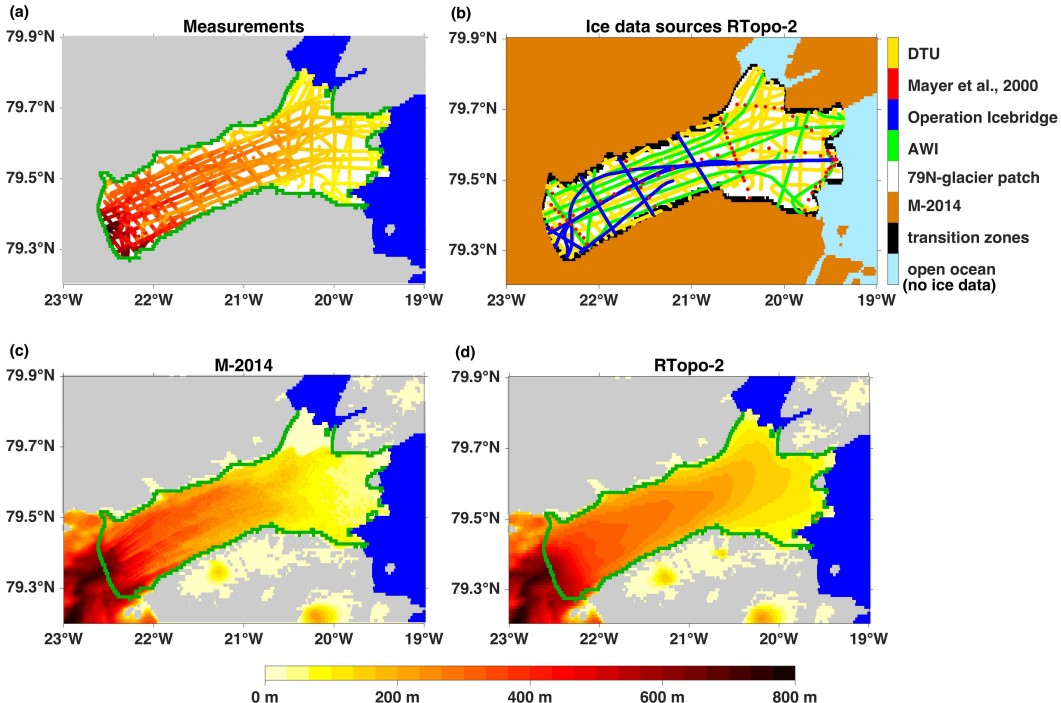

**Figure 5.** Ice thickness maps of the floating ice tongue of Nioghalvfjerdsfjorden Glacier. The maps show the coverage of ice thickness measurements from radar and seismic soundings (a), data sources used in RTopo-2 (b), and ice thicknesses in M-2014 (c) and RTopo-2 (d). Shaded in grey is land, blue shaded is the open ocean. The dark green line in panels (a), (c) and (d) indicates the grounding line.

### 2.3.2 Northeast Greenland glacier topographies

Given that the floating ice tongue of Nioghalvfjerdsfjorden Glacier is one of the very few places in the Arctic where ice and ocean interact at an ice shelf base that covers more than just a very small area, this is a region of particular scientific interest. We therefore decided to enhance the ice thickness data in this area by using recently obtained airborne radar data as well as

5  seismic soundings (Mayer et al., 2000) and airborne data from 1998, (Fig. 5). We used spherical triangulation to interpolated the ice thickness data in the area of the floating ice tongues of Nioghalvfjerdsfjorden Glacier and Zachariæ Isstrøm to our regular grid. The resulting ice thickness map was smoothed along the flow lines to avoid interpolation artefacts. Compared to M-2014, the main benefit of our grid is that it also covers the thickness of floating ice in Dijmphna Sund (Figs. 5c and 5d).

Assuming hydrostatic equilibrium we calculated the surface height ($\zeta$) of floating ice from the gridded ice thickness ($H$)

10  using

$$\zeta = H \frac{\rho_{water} - \rho_{ice}}{\rho_{water}} \tag{1}$$

with densities ($\rho$) of $\rho_{water} = 1023 \, \mathrm{kg \, m^{-3}}$ and $\rho_{ice} = 917 \, \mathrm{kg \, m^{-3}}$. The ice draft results from $\zeta - H$.



We combined the newly gridded glacier topographies with the surrounding surface height and ice draft maps with a transition zone of 2 km width. Corrections needed to be applied in areas where the newly gridded ice thickness exceeded the water depth. In regions where the surface type mask derived from M-2014 proposes the existence of floating ice, bedrock topography was corrected by applying a minimum water column thickness of 1 m. This procedure is justified by the fact that ice thickness observations for the floating ice tongues are much more densely spaced than the very sparse sub-ice bathymetry measurements obtained from seismics.

In comparison, the RTopo-2 ice thickness map derived from measurements deviates from M-2014 mostly towards the glacier front (Figs. 5c and 5d). East of 20.5°W, the ice is up to 80 m thicker in RTopo-2. For the part of the glacier front which extends northward into Dijmphna Sund the ice thickness in M-2014 is only 1 m with a classification as grounded ice. In contrast, based on the observations from, e.g., Mayer et al. (2000), RTopo-2 shows floating ice with thicknesses up to 150 m in the same area.

### 2.3.3 Antarctic ice and bedrock topographies

We used the bedrock topography from IBCSO Version 1.0 (Arndt et al., 2013) (polar stereographic grid with true scale at 65° S) for the bathymetry of the Southern Ocean including the sub-ice shelf cavities. In the north, a 50 km wide transition zone along 61.5° S connects the IBCSO data to the GEBCO_2014 grid. On the Antarctic continent, bedrock topography is derived from the Bedmap2 (Fretwell et al., 2013) data set, as are the surface and ice bottom topographies.

Where the coastline of the Antarctic continent is formed by a transition from grounded ice or bare land to open ocean, we join the Bedmap2 and IBCSO topographies in a narrow band directly at the coast. Along the grounding lines of sub-ice cavities, the transition between the IBCSO and Bedmap2 topographies is in a roughly 8 km wide band 10 km off the grounding line (i.e. within the sub-ice cavity). In any case, a smooth transition between the IBCSO and Bedmap2 grids is easy to ensure due to the fact that Bedmap2 bedrock topography data have been incorporated in the generation of IBCSO. Small inconsistencies that still arise from the interpolation (mainly due to the discontinuity of ice draft along the ice front) were cured by enforcing grounded ice bottom topography to be identical to bedrock topography.

Given that the IBCSO data set incorporates not only bedrock relief but also ice surface topography from Bedmap2, it may seem better to use the IBCSO products throughout Antarctica and thus avoid the stitching between the two grids along the Antarctic continent. We decided not to follow this approach because IBCSO does not provide information about the thickness of floating ice shelves. Given that the compilation of RTopo-2 has been targeted towards studies of ice dynamics and ice-ocean interaction at the interfaces between ice sheets and ocean, we decided that discontinuities of ice thickness across the grounding lines are to be avoided as far as possible. Therefore, ice surface and bottom topographies for grounded and floating ice are to be adopted from one self-consistent data set, which is possible only with Bedmap2. Similar consistency arguments apply to the bedrock relief under grounded ice; again we decided to use the original Bedmap2 product here to avoid introducing inconsistencies.



### 2.3.4   Local corrections for Antarctic sub-ice shelf bathymetry

For Filchner-Ronne Ice Shelf and the ice streams in its catchment basin, as well as for the ice topographies in many other regions, the benefit of a largely improved data coverage and resolution in Bedmap2 is very obvious and quite substantial. However, with regard to the representation of sub-ice cavity bathymetry, the transition from RTopo-1 to Bedmap2 does not

universally yield an improvement. Although Fig. 6 in Fretwell et al. (2013) indicates that sub-ice shelf bathymetry for most of the ice shelves in Bedmap2 goes back to RTopo-1, many details of cavity bathymetry that appear plausible and are in some cases well covered by original data have vanished in the transition. This section reports on the local data corrections or reconstruction procedures we applied.

**Getz and Abbot Ice Shelves**

According to Fretwell et al. (2013), sub-ice bathymetry for Getz Ice Shelf cavity in Bedmap2 (Fig. 6a) has been derived from the topography grid of Nitsche et al. (2007). While this data set provides an excellent bathymetry map for the open Amundsen Sea, it suffers from missing data for the sub-ice shelf cavities. As a result, Bedmap-2 suggests a very shallow water column in large parts of the cavity. For RTopo-2 (Fig. 6b), we decided to go back to the submarine trough structure that RTopo-1 in-

herited from ALBMAP (Le Brocq et al., 2010). Upper and lower ice surface topographies and the surface type mask (locations of coast and grounding line) continue to be derived from Bedmap2. A smooth transition of sub-ice shelf bathymetry to the bedrock topography under grounded ice (from Bedmap2) and to open-ocean bathymetry (from IBCSO) is achieved using tanh functions in blending zones of $\approx 15$ km width (Fig. 6c). As a result, water column thickness in the sub-ice cavity (Fig. 6d) features continuous troughs extending from the open-ocean continental shelf across the ice front towards the grounding line.

The high basal melt rates suggested for Getz Ice Shelf (e.g., Depoorter et al. (2013)) makes the existence of such transport pathways for warm water seem plausible. A strict evaluation, however, is made very difficult by the lack of sub-ice bathymetry data, and there is no proof that the structures we suggest are correct.

A similar case can be made for Abbot Ice Shelf. For the eastern part of Abbot Ice Shelf, sub-ice bathymetry in Bedmap2 is derived from the Graham et al. (2011) data set, which incorporates ALBMAP bedrock topography in the sub-ice cavity. For

the western part of the Abbot Ice Shelf, however, Bedmap2 utilizes the bathymetry map of Nitsche et al. (2007), which again leads to a very small water column thickness with virtually no connection to the open ocean in this sector of the ice shelf. We decided to restore the structure of a sub-ice trough connected to the eastern Amundsen Sea from ALBMAP for the Abbot Ice Shelf cavity west of 98° W. Also here, it should be kept in mind that bathymetry under this ice shelf is only weakly constrained.






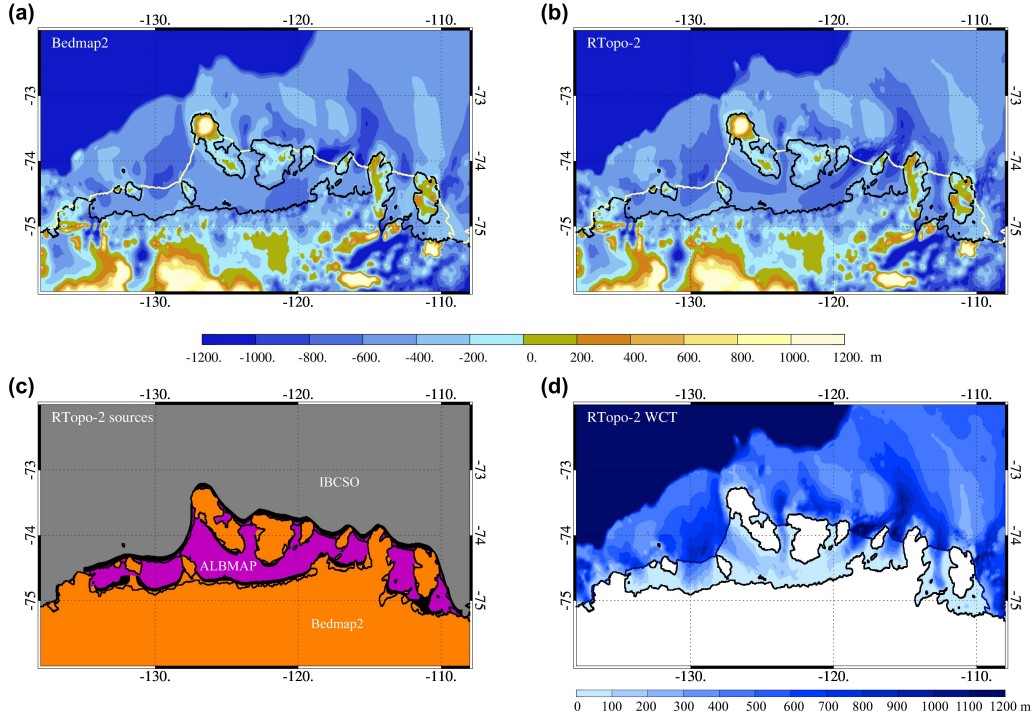

**Figure 6.** Top line: Bathymetry for Getz Ice Shelf cavity and its surrounding in Bedmap2 (a) and the merged bathymetry in RTopo-2 (b). Panel (c): Indication of data sources for the RTopo-2 bathymetry grid, with colors corresponding to the global source map in Fig. 2. Panel (d): Water column thickness obtained from RTopo-2 ice shelf draft and bathymetry.

**Larsen C Ice Shelf**

For Larsen C Ice Shelf cavity, Fig. 6 in Fretwell et al. (2013) indicates that bedrock topography was derived from RTopo-1. On a closer look, however, substantial differences between Bedmap2 and RTopo-1 can be seen in this area. Specifically, many of the deep troughs that had been inferred from ice draft at the grounding line in RTopo-1 have been removed (or are far less pronounced) in Bedmap2. Revisiting all the data sets in question here, we found that the RTopo-1 ice and bedrock topographies along the southern part of Larsen C grounding line imply an ice thickness maximum to be found not within the ice streams feeding the ice shelf but shortly downstream from where the ice comes afloat. This pattern - it is most obvious for the ice stream between the Joerg and Kenyon Peninsulas in the southwestern corner of Larsen C Ice Shelf - clearly seems not very plausible. Given that it can be safely assumed that the ice topographies in Bedmap2 are more reliable than the combination of data sets used in RTopo-1, we conclude that there is no sufficient evidence for the existence of the deep throughs suggested near the Larsen C grounding line in RTopo-1. RTopo-2 thus simply adopts the Larsen C cavity geometry from Bedmap2.

**Fimbulisen**

Bedmap2 bathymetry under the floating Fimbulisen is claimed to be derived from RTopo-1 but in fact deviates from the lat-



ter substantially. Specifically, the deep troughs between the islands (ice rises) in the eastern part of the cavity (i.e. between 1° E and 5° E) that Nøst (2004) infered from original seismic data are no longer there. We decided to go back to the Nøst (2004)/Smedsrud et al. (2006) data set that was already incorporated in RTopo-1 for the Fimbulisen cavity between 1° E and 5° E.

## 2.4   Tabular iceberg in Weddell Sea

In August 1986, three giant icebergs (A-22, A-23, and A-24), each one between 3000 and 4000 km$^2$ in area, calved from the Filchner Ice Shelf front (Ferrigno and Gould, 1987) and grounded at the eastern Berkner Bank. Iceberg A-24 came ungrounded in March 1990 and drifted northward through the Weddell and Scotia Seas; icebergs A-22 and A-23 broke in two in 1994 and 1991, respectively. One of their remnants, A-23A, is still grounded on the eastern slope of Berkner Bank and continues to form a barrier to the sea ice drifting in this region, frequently creating a polynya in its lee (Markus, 1996). When modelling sea ice in this area, a comparison between modeled and observed (mostly remote sensing) data is strongly complicated if this effect is omitted in the model (see, e.g. Haid and Timmermann (2013)). A similar case can be made for the iceberg's effect on the ocean currents on the continental shelf. Despite the fact that Grosfeld et al. (2001) showed that iceberg calving and grounding does change the circulation and hydrography in the Filchner ice shelf-ocean system, it is not common for today's ocean models to take this into account.

To enable high-resolution modelers to do the model-to-data comparison in a more consistent way - especially given that data coverage is about to improve considerably in the framework of ongoing and planned field activities in the area - and to achieve a more realistic representation of the local ocean currents in hindcast simulations, we decided to include the signature of A-23A in RTopo-2. The area covered by the iceberg was picked from a composit of 2013 MODIS images (Paul et al., 2015) and defined to be covered with grounded ice, i.e. ice with a lower surface topography identical to the ocean bathymetry (which clearly is only a schematic representation of the real situation). Freeboard of the iceberg is represented as a constant value in RTopo's surface height field; it is computed from Archimedes' principle assuming densities of ocean and ice to be 1028 and 910 kg/m$^3$, respectively. Thickness of the iceberg in this equation is assumed to be such that its draft is equal to the minimum water depth in the area that is covered by the iceberg. Eventually, this procedure yields a freeboard (surface height) value of about 42 m over the iceberg area, which is consistent with the freeboard derived from SAR interferometry applied to TanDEM-X image pairs of June 2013 (M. Rankl, pers. comm. 2016).

Note that the original bathymetry grid under the iceberg is fully preserved so that the whole feature can be removed without any loss of information in case this seems desirable in any particular application of the data set.

## 2.5   Surface type mask

In addition to the maps of the bedrock and ice topographies we provide a global mask which distinguishes between open ocean, bare land, grounded ice and floating ice (Fig. 7). On the polar continents, the mask largely follows M-2014 for Greenland and Bedmap2 for Antarctica. Ice caps not connected to the Antarctic ice sheet or Greenland main land have been removed from the



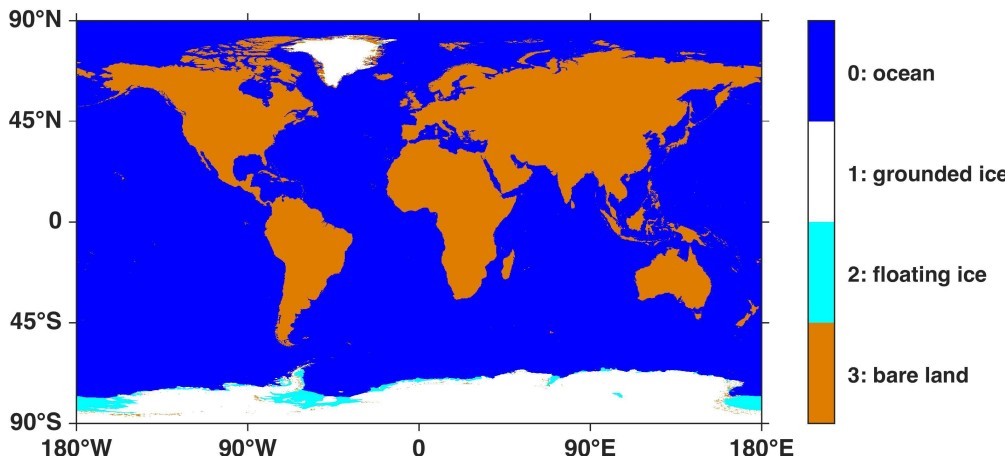

**Figure 7.** Global surface type mask (compare with Timmermann et al. (2010), Fig. 7 but Greenland ice sheet included).

mask and classified as bare land. Ice surface height in these cases has been adopted as bedrock surface height. In contrast to RTopo-1, the surface type mask in RTopo-2 contains information about rock outcrops (surface type 'bare land') in Antarctica.

Lakes and enclosed seas outside Antarctica and Greenland are marked as bare land in the mask, but are still present in the bathymetry data set adopted from GEBCO_2014. This was done to avoid the tedious procedure of manually removing features

with a topography below sea level and no connection to the world ocean when setting up an ocean general circulation model.

For the Northeast Greenland continental shelf, the NEG_DBM bathymetry map provides bedrock elevation with a very high data coverage. We used this data set to adjust the surface type mask: Grid points with an elevation > 0 were classified as bare land surface, negative elevation obviously enforced a classification as ocean. Within the perimeter of the Greenland continent, the surface type mask remained being defined by M-2014.

## 3  Error estimates

In the transition corridors between different data sets and in regions where values have been inferred from consistency arguments errors are hard to quantify. Here we can only give an overview on error estimates provided by the authors of some of our source data sets.

### 3.1  Bathymetry

In GEBCO_14 approximately 18% of the non-land grid cells are constrained by bathymetric control data, which consists of echosounding data as well as pre-prepared bathymetric grids that may contain interpolated areas (Weatherall et al., 2015). In the current IBCAO version, the Arctic Ocean is mapped by multibeam surveys covering 11% of the area and an additional





vast amount of single beam data (Jakobsson et al., 2012). In IBCSO around 17% of grid cells in the Southern Ocean are directly constrained with data; 15.4% of data points are from multibeam bathymetry (Arndt et al., 2013). In areas with no direct measurements, the ocean bathymetry was interpolated between measurements and/or plausible bathymetry.

In general, the accuracy of echosounding systems can be expected to be about one percent of the water depth. However, in
the areas between the sounding tracks uncertainties can be much higher.

## 3.2   Ice and bedrock topographies

For ice surface heights of Greenland the overall root-mean-square deviation between the GIMP digital elevation model and ICESat elevation is ± 9.1 m (Howat et al., 2014). The deviation varies strongly with region (Fig. 6 in Howat et al. (2014)).

The technical error of ice thicknesses derived from radio echosounding depends mainly on the sampling interval and trans-
mitted signal length, which both varies from system to system. The resolution in ice thickness of the various employed RES systems varies between 5.05 and 8.45 m; the sampling precision is higher, usually in the order of 1 m. Thus an uncertainty of about 15 to 35 m for the ice thickness is realistic. However complex geometries and steep topography confining the investigated glaciers and ice tongues can cause side and multiple reflections which mask the subglacial reflections, especially in airborne measurements.

Ice thickness and ice draft mapped by Morlighem et al. (2014) are subject to errors of about 35 m for areas with a dense radar sounding coverage. In areas which are less well constrained, errors can exceed 50 m (Morlighem et al., 2014).

Ice thickness maps derived from the available observations for Nioghalvfjerdsfjorden Glacier and Zachariæ Isstrøm reveal distinct differences between data sets from different years. All data across Zachariæ Isstrøm are based on radar data from 2010 to 2014 (obtained from Operation Icebridge and AWI flights). Based on Landsat optical imagery, Mouginot et al. (2015)
observed an accelerated retreat in the ice front position in 2013/14 and estimated a mass loss of 5 Gt yr$^{-1}$. The extent of Zachariæ Isstrøm in RTopo-2 thus represents the state prior to its decay and not the present state.

Ice thickness data covering Nioghalvfjerdsfjorden Glacier include additionally a large number of radar tracks and seismic data obtained ten years earlier in 1997/98 (DTU / Seroussi et al. (2011), Mayer et al. (2000)) (see Fig. 5b). Data from the two time slices differ by about 50 m in several places, especially within 5 km from the grounding line where the floating ice tongue
is subject to strong basal melt. The differences are in the same range as the along-track noise in some of the radar tracks (see above). Such discrepancies were smoothed out by our interpolation procedure.

Next to the uncertainties related to data interpretation and processing, the representation of the firn layer ('firn correction') is an issue that requires serious attention. While in B-2013 a firn layer thickness of 10 m in all ablation regions around Greenland is assumed, there is no firn correction applied in M-2014. Snow depth varies strongly over the Greenland Ice Sheet and within
the seasonal cycle (Nghiem et al., 2005), with most of the snow melting during the summer season close to the coast. A constant-value firn correction is therefore bound to be a rather crude approximation. We keep the M-2014 assumption of zero firn layer thickness and note that for determining a regionally varying firn correction the local depth-density relationship, respectively the variation of the dielectric properties with depth needs to be known.





## 4 Summary and outlook

We compiled a global 30-arc second data set for World Ocean bathymetry and Greenland/Antarctic ice sheet/shelf topography. High-resolution data from Greenland floating glaciers and of bathymetry on the northeast Greenland continental shelf were compiled into a synthesis of gridded bathymetry products including the Morlighem et al. (2014) Greenland ice and bedrock
topographies and the Bedmap2 Antarctic topography data sets. Similar to RTopo-1 (Timmermann et al., 2010), the RTopo-2 data set contains maps of global bedrock topography and the upper and lower surface heights of the Antarctic and Greenland ice sheet/ice shelf system. Consistent with the topography maps, a surface type mask for open ocean, grounded ice, floating ice, and bare land surface is provided.

This new data set provides enough local detail for a wide range of global or regional studies. Our main target group are
ocean modelers who aim at a realistic representation of ice-ocean interaction in an ocean general circulation or climate model. In the current version, particular attention has been paid to the floating glaciers and the continental shelf in the northeast Greenland sector. Other Greenland fjord regions are of similar interest but suffer from a lack of data. We encourage users who are specifically interested in one of those fjords to carefully review the data using information unused by us as a benchmark. Additional contributions of (gridded or ungridded) fjord/shelf bathymetry and/or glacier/ice shelf/cavity geometry are welcome
and will be used to update the data set as soon as possible.

## 5 Data access

The RTopo-2 data set is available in NetCDF format in four different variations at doi:10.1594/PANGAEA.856844:

1. The complete global 30-arc second data set has been split into four files:

    – RTopo-2.0_30sec_bedrock_topography.nc (3.73 GB),
– RTopo-2.0_30sec_ice_base_topography.nc (3.73 GB),
    – RTopo-2.0_30sec_surface_elevation.nc (3.73 GB),
    – RTopo-2.0_30sec_aux.nc (2.8 GB) which contains auxiliary maps for data sources and the surface type mask.

2. A regional 30-arc second subset that covers all variables around Greenland in the interval 80°E - 0°, 55 - 85 °N is available in:

– RTopo-2.0_30sec_Greenland.nc (518.7 MB).

3. A regional 30-arc second subset for the Antarctic region south of 50°S has been split into two files:

    – RTopo-2.0_30sec_Antarctica_data.nc (2.49 GB) contains bedrock topography, ice base topography, and surface elevation,
    – RTopo-2.0_30sec_Antarctica_aux.nc (622.3 MB) contains auxiliary maps for data sources and the surface type
mask.




4. A complete global 1-arc minute data set that has been split into two files:

   – RTopo-2.0_1min_data.nc (2.8 GB) contains maps of bedrock topography, ice bottom topography, and surface elevation,

   – RTopo-2.0_1min_aux.nc (700.1 MB) contains auxiliary maps for data sources and the surface type mask.

5 Data sets for the location of coastlines (RTopo-2.0_coast.asc, 53.4 MB) and the ice shelf/floating glacier front lines (RTopo-2.0_isf.asc, 1.7 MB) have been prepared in ASCII format. Grounding lines are represented as parts of the coastline. To enable communication in case of errors or updates, we would appreciate a notification from users of our data set.

*Author contributions.* R. Timmermann (Southern hemisphere) and J. Schaffer (Northern hemisphere) designed the merging strategies and processed the RTopo-2 data sets. J.E. Arndt and M. Morlighem provided the latest versions of bathymetry and ice thickness maps for Greenland and the continental shelf in its vicinity. S.S. Kristensen, C. Mayer and D. Steinhage provided pre-processed ice thickness data for Nioghalvfjerdsfjorden Glacier. J. Schaffer prepared the manuscript with contributions from all co-authors.

*Acknowledgements.* The authors would like to thank S. Paul and R. Zentek for extracting the iceberg position from the MODIS data, S. Coers, B.K. Galton-Fenzi, H. Gudmundsson, H.H. Hellmer, D. Jansen, and L. Padman for helpful discussions, and W. Cohrs, H. Liegmahl-Pieper, and C. Wübber for providing excellent computing facilities at AWI. GEBCO_2014 Grid (version 20150318) data were obtained from http://www.gebco.net. Some data used in this paper were acquired by NASA's Operation IceBridge project. Funding by the Helmholtz Climate Initiative REKLIM (Regional Climate Change), a joint research project of the Helmholtz Association of German research centres (HGF) is gratefully acknowledged.



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
