# Peer review of "A global high-resolution data set of ice sheet topography, cavity geometry and ocean bathymetry"

_Earth System Science Data, 2016_

## Referee Comment (RC1) · Anonymous Referee #1 · 14 Jul 2016

Review for "A global high-resolution data set of ice sheet topography, cavity geometry and ocean bathymetry"
By: Schaffer *et al*. (ESSD Discussion: 8 June 2016)

The authors present a comprehensive, global dataset containing bathymetry and ice surface and basal topography for the Greenland and Antarctic ice sheets. This work is based on and expands a previous effort to define a global dataset aimed primarily at ocean and climate modelers.

My overall impression of the manuscript is very positive and I think it should be published following some minor revisions outlined below. The manuscript is mostly well written, cogent, concise, and logically organized. The figures are informative and well presented. My biggest critique in the manuscript is that the authors should provide more detail on the procedure used to smoothly transition between different datasets. There are a number of ways to smoothly blend datasets and it would be nice to see a few sentences devoted to describing the chose method(s).

The dataset is accessible via the link provided in the manuscript. The data appear to be of good quality and are easy to use in QGIS. It is likely that users of RTopo-1 (the precursor to these data) and others will find these data useful.

General comments:
1. In general, what the authors are calling resolution is actually grid spacing. While it is common in GIS applications to treat the terms resolution and grid spacing as synonyms, there is an important distinction between the two and a precise discussion of datasets should recognize that distinction. Resolution is the smallest scale that is observable in a given measurement. In other words, resolution is a statement of information content, and therefore should not be applied to interpolated grids, except in special cases and with careful qualifications. Grid spacing, on the other hand, is simply part of the metadata for a gridded dataset and says nothing explicit about information content.
2. Did the authors develop the tools needed to interpolate source data to a common grid or use existing tools (ArcGIS, GDAL, etc.)? If existing tools were used, please reference where appropriate.

Section comments:
1. The dataset contains geometry for both Greenland and Antarctica but the Introduction only discusses Greenland. A second paragraph discussing Antarctica and the importance of sub-ice-shelf cavity geometry should be added.
2. Subsection partitioning in Section 2 (Datasets and processing) gets a little confusing after the 3rd subsection. I suggest getting rid of the first subsection (not the content, just the section numbering) and treating each major region (e.g. oceans, Greenland, Antarctica) as the first subsection level. In other words, what is now 2.1 would simply be the summary paragraph for section

2; what is now 2.2.1 would be 2.1, and so on. Please number all subsections if allowed by the journal format.

Minor and grammatical corrections (numbers given as page.line):

Title: Original title has a few grammatical errors and, without changing the wording, should read like: "A global, high-resolution dataset of ice-sheet topography, cavity geometry, and ocean bathymetry" (Oxford common is optional).

Everywhere: "Data set" can be written as a single word.

Everywhere: Distance descriptors of the form X-arc-minutes should contain a hyphen between "arc" and the fractional degree unit.

1.6. "with now 30-arc seconds resolution" should read "with 30-arc-second grid spacing"

2.4. cavity -> cavities

3.6. Suggest rewording the sentence to read: Surface lakes were deleted (give brief reasoning for this) and are presented as bare land. Subglacial lakes in Antarctica have been preserved.

3.11. The last sentence ("Interpolation of the source datasets…") is vague. Please provide a reference or be more explicit as to which triangulation approach was used, where artifacts come from, and what the authors mean by "careful smoothing."

4.2. "…local expertise of regional undersea mapping projects" should read "…information from regional undersea mapping projects"

4.13. "smooth blending" is too vague

7.14. Please describe in more detail (or reference) how datasets are combined in the transition zones.

9.21. "As already mentioned in…" could be reworded to be "As discussed in…"

10.5. interpolated -> interpolate

13.7. Sentence beginning with "This pattern…" should be reworded to be less awkward.

16.10. "…which both varies…" should read "…both of which vary…"

16.10. In the sentence beginning "The resolution in ice thickness," it seems like the authors are referring to vertical resolution (as opposed to horizontal resolution in much of the rest of the manuscript). Please clarify.

---

## Referee Comment (RC2) · M. Jakobsson (Referee) · 23 Jul 2016

Overall quality: This paper presents a new global gridded compilation that contains bathymetry, upper and lower ice surface topographies and global surface height. This follows from the RTopo-1 compilation and the resulting new compilation is released under the name of RTopo-2. This is an important contribution which most certainly will be useful for broad range of geoscientists in the need of a coherent global dataset with merged bathymetry and under-ice topography. In addition, an updated useful global surface type mask is provided. For these reasons alone, I believe the paper warrants publication. Beyond discussing the scientific needs for the kind of gridded data compilation in focus, "data release papers" of this kind serve a couple particularly important points: 1) presenting the data sources and compilation methods 2) discuss errors and data limitations 3) making it possible for the data users to cite a peer-review

publication. Do I believe this paper fulfill these points? Yes, but it could be improved on some aspects. In particular, I miss a more technical and detailed description on the compilation methods used. What software/algorithms were involved? Flow charts of the procedures would be much welcomed. This information could be presented in supplementary material if it proves difficult to include in the main paper, as long as it is accessible for the reader somehow.

Specific comments: The main component missing in this paper is a section including the technical details on the compilation procedures. I propose adding such a section. For example, it is now written on page three under the main section Data sets and processing: "Interpolation of the source data sets from their different projections to our geographic grid was done by triangulation; a careful smoothing was applied to avoid artefacts." How was this done? One can go from a TIN to a grid in many ways. What is a careful smoothing? Was the smoothing applied all over the resulting surface? What algorithms were used, and in what software?

I believe it is not made crystal clear that this is "RTopo-2" in the abstract, it is just written briefly and assumed that the reader understands. Perhaps something like: "We followed the spirit of the global RTopo-1 data set and compiled an update referred to as RTopo-2 using consistent maps of global ocean bathymetry, upper and lower ice surface topographies and global surface height. RTopo-2 is comprised of a spherical grid with 30-arc seconds cell spacing. . . . . .." (I suggest using cell-spacing rather than "resolution" since the latter may confuse some that source data actually exist consistently on the resolution of 30-arc seconds).

The bathymetry near the Greenland coast, specifically in the fjords, is extremely poorly constrained even if a few more data sets are included in this compilation. I think this should be stronger high-lighted in the main paper. On the same subject it is written in the abstract: "In particular, we aimed at a good representation of the fjord and shelf bathymetry surrounding the Greenland continent." This is fine, but it is followed by "We corrected data from earlier gridded products in the areas of Petermann Glacier, HagenBræ and Sermilik Fjord assuming that sub-ice and fjord bathymetries roughly follow plausible Last Glacial Maximum ice flow patterns." Even if this certainly is an improvement likely closer to the truth, it is another assumption not based on real bathymetric data. For this reason, I do not think it is appropriate referring to it as a "correction". Instead "modified" is more appropriate. Could this modification be shown graphically?It is not made fully clear how this assumption is technically making its way into the gridded compilation.

On page 6, please clarify the last part of the sentence: …."each other; combining surface elevation and ice thickness maps yields an ice bottom topography that even for grounded ice is not identical to the bedrock topography grid provided.". . .

On page 2 it is written: "This is particularly true for the northeast Greenland continental shelf, where a system of troughs provides a flux of warm water towards the floating ice tongues of Nioghalvfjerdsfjorden Glacier (also referred to as 79 North Glacier) and Zachariæ Isstrøm (Arndt et al. (2015), Wilson and Straneo (2015)). "

Why is this particularly true here, troughs of western Greenland seems equally pronounced and the same problem apply? We know less about the coupling between the fjords and the glacial troughs along the northwester sector of the Greenland coast, north of Upernavik), but this does not exclude them as being sensitive from influx of warmer subsurface water.
* * *

---

## Author Comment (AC1) · 5 Oct 2016

"Dear reviewer, thank you so much for your careful reading and helpful comments. You really helped to improve the manuscript! Below you find your comments and our responses (in quotation marks). Regarding our changes you find a revised marked-up manuscript in the supplement. Best regards, Janin Schaffer & Ralph Timmermann"

Review for "A global high-resolution data set of ice sheet topography, cavity geometry and ocean bathymetry" By: Schaffer et al. (ESSD Discussion: 8 June 2016) The authors present a comprehensive, global dataset containing bathymetry and ice surface and basal topography for the Greenland and Antarctic ice sheets. This work is based on and expands a previous effort to define a global dataset aimed primarily at ocean and climate modelers. My overall impression of the manuscript is very positive

and I think it should be published following some minor revisions outlined below. The manuscript is mostly well written, cogent, concise, and logically organized. The figures are informative and well presented. My biggest critique in the manuscript is that the authors should provide more detail on the procedure used to smoothly transition between different datasets. There are a number of ways to smoothly blend datasets and it would be nice to see a few sentences devoted to describing the chose method(s). The dataset is accessible via the link provided in the manuscript. The data appear to be of good quality and are easy to use in QGIS. It is likely that users of RTopo-1 (the precursor to these data) and others will find these data useful.

General comments:

1. In general, what the authors are calling resolution is actually grid spacing. While it is common in GIS applications to treat the terms resolution and grid spacing as synonyms, there is an important distinction between the two and a precise discussion of datasets should recognize that distinction. Resolution is the smallest scale that is observable in a given measurement. In other words, resolution is a statement of information content, and therefore should not be applied to interpolated grids, except in special cases and with careful qualifications. Grid spacing, on the other hand, is simply part of the metadata for a gridded dataset and says nothing explicit about information content.

- "We agree. Changes were applied accordingly throughout the whole manuscript."

2. Did the authors develop the tools needed to interpolate source data to a common grid or use existing tools (ArcGIS, GDAL, etc.)? If existing tools were used, please reference where appropriate.

- "We developed our own tools to interpolate source data to a common grid. We describe this in more detail in the revised manuscript (section 2.2)."

Section comments:
1. The dataset contains geometry for both Greenland and Antarctica but the Introduction only discusses Greenland. A second paragraph discussing Antarctica and the importance of sub-ice-shelf cavity geometry should be added.

- "We agree. A paragraph about Antarctica has been added to the Introduction."

2. Subsection partitioning in Section 2 (Datasets and processing) gets a little confusing after the 3rd subsection. I suggest getting rid of the first subsection (not the content, just the section numbering) and treating each major region (e.g. oceans, Greenland, Antarctica) as the first subsection level. In other words, what is now 2.1 would simply be the summary paragraph for section 2; what is now 2.2.1 would be 2.1, and so on. Please number all subsections if allowed by the journal format.

- "We restructured and slightly changed the subheadings (bold) of section 2 as follows: 2.1 Overview of RTopo-2 maps 2.2 Data sources and merging procedure 2.3 Bedrock and bathymetry data sets 2.4 Ice sheet topography and cavity geometry 2.5 Tabular iceberg in Weddell Sea We kept the subsubsections but they now relate clearer to the subsections. The discussion of cavity bathymetry for 79N glacier has been moved from 2.3.4 to 2.4.2 to ensure consistency with the new headings."

Minor and grammatical corrections (numbers given as page.line):

Title: Original title has a few grammatical errors and, without changing the wording, should read like: "A global, high-resolution dataset of ice-sheet topography, cavity geometry, and ocean bathymetry" (Oxford common is optional). - "Changes applied."

Everywhere: "Data set" can be written as a single word. - "We stick to the term data set, which we already used for our RTopo-1 publication. No change applied."

Everywhere: Distance descriptors of the form X-arc-minutes should contain a hyphen between "arc" and the fractional degree unit. - "Changes applied."

1.6. "with now 30-arc seconds resolution" should read "with 30-arc-second grid spacing" - "Change applied throughout the whole manuscript."
2.4. cavity -> cavities - "Change applied."

3.6. Suggest rewording the sentence to read: Surface lakes were deleted (give brief reasoning for this) and are presented as bare land. Subglacial lakes in Antarctica have been preserved. - "We realized that the statement was not quite accurate anyway: The depth signature of surface lakes from GEBCO_2014 is nicely preserved in the 'bathymetry/bedrock' field we provide. The surface type mask, however, still indicates them as 'bare land'. We discuss this (with the full information and a bit of reasoning) in Section 2.6 ("Surface type mask"), so we decided to remove the misleading sentence here."

3.11. The last sentence ("Interpolation of the source datasets...") is vague. Please provide a reference or be more explicit as to which triangulation approach was used, where artifacts come from, and what the authors mean by "careful smoothing." - "We expanded this last sentence into a new section (2.2) that gives substantially more details."

4.2. "...local expertise of regional undersea mapping projects" should read "...information from regional undersea mapping projects" - "Change applied."

4.13. "smooth blending" is too vague - "We added a statement towards the use of tanh functions."

7.14. Please describe in more detail (or reference) how datasets are combined in the transition zones. - "Again, this is covered in the "Data sources and merging strategies" section 2.2 of the revised manuscript; the important information here is the width of the transition zone."

9.21. "As already mentioned in..." could be reworded to be "As discussed in..." - "Change applied."

10.5. interpolated -> interpolate - "Change applied."

13.7. Sentence beginning with "This pattern..." should be reworded to be less awkward.

- "We tried our very best (see revised manuscript 14.30)."

16.10. "...which both varies..." should read "...both of which vary..." - "Change applied."

16.10. In the sentence beginning "The resolution in ice thickness," it seems like the authors are referring to vertical resolution (as opposed to horizontal resolution in much of the rest of the manuscript). Please clarify. - "Here we indeed refer to the vertical resolution. Change applied."

Please also note the supplement to this comment:
http://www.earth-syst-sci-data-discuss.net/essd-2016-3/essd-2016-3-AC1-supplement.pdf

---

## Author Comment (AC2) · 5 Oct 2016

"Dear Dr. Jakobsson,

thank you very much for careful reading and your truly helpful comments! Below you find your comments and our responses (in quotation marks). Regarding our changes you find a revised marked-up manuscript in the supplement.

Best regards, Janin Schaffer & Ralph Timmermann"
and lower ice surface topographies and global surface height. This follows from the RTopo-1 compilation and the resulting new compilation is released under the name of RTopo-2. This is an important contribution which most certainly will be useful for broad range of geoscientists in the need of a coherent global dataset with merged bathymetry and under-ice topography. In addition, an updated useful global surface type mask is provided. For these reasons alone, I believe the paper warrants publication. Beyond discussing the scientific needs for the kind of gridded data compilation in focus, "data release papers" of this kind serve a couple particularly important points: 1) presenting the data sources and compilation methods 2) discuss errors and data limitations 3) making it possible for the data users to cite a peer-review publication. Do I believe this paper fulfill these points? Yes, but it could be improved on some aspects. In particular, I miss a more technical and detailed description on the compilation methods used. What software/algorithms were involved? Flow charts of the procedures would be much welcomed. This information could be presented in supplementary material if it proves difficult to include in the main paper, as long as it is accessible for the reader somehow.

- "We have added information about the software used and the algorithm applied in the new section 2.2.

Flowcharts are not straightforward to provide here, because the data sets used for RTopo-2 have little in common; they all differ in terms of the variables covered and uncertainties. The choices we had to make are mostly about what to do in order to ensure consistency – do we trust the bedrock and adjust the thickness of grounded ice? Vice versa? Do we trust both equally? Or one slightly more than the other? Do we go for locally exact rendition of source data or for a global map with continuity and consistency valued over exactness? (Note that "exact" does not mean "correct" for many places). These are choices that could not be made in an objective way; they are indeed largely subject to an "educated intuition" of the first two authors of the paper and were largely guided by the question which choice would entail the smallest risk of

causing damage to an ice or ocean model. So, we decided not to come up with an overly schematic view of what we did. Instead, for each of the data sets included, we tried to report on the choices we made and the reasoning behind them."

Specific comments:

The main component missing in this paper is a section including the technical details on the compilation procedures. I propose adding such a section. For example, it is now written on page three under the main section Data sets and processing: "Interpolation of the source data sets from their different projections to our geographic grid was done by triangulation; a careful smoothing was applied to avoid artefacts." How was this done? One can go from a TIN to a grid in many ways. What is a careful smoothing? Was the smoothing applied all over the resulting surface? What algorithms were used, and in what software?

- "We expanded that sentence into new section 2.2, providing much more detail than before."

I believe it is not made crystal clear that this is "RTopo-2" in the abstract, it is just written briefly and assumed that the reader understands. Perhaps something like: "We followed the spirit of the global RTopo-1 data set and compiled an update referred to as RTopo-2 using consistent maps of global ocean bathymetry, upper and lower ice surface topographies and global surface height. RTopo-2 is comprised of a spherical grid with 30-arc seconds cell spacing. . .. . .." (I suggest using cell-spacing rather than "resolution" since the latter may confuse some that source data actually exist consistently on the resolution of 30-arc seconds).

- "We modified the way "RTopo-2" is introduced in the abstract. We use the term grid spacing instead of resolution throughout the whole manuscript now."

The bathymetry near the Greenland coast, specifically in the fjords, is extremely poorly constrained even if a few more data sets are included in this compilation. I think this

should be stronger high-lighted in the main paper. On the same subject it is written in the abstract: "In particular, we aimed at a good representation of the fjord and shelf bathymetry surrounding the Greenland continent." This is fine, but it is followed by "We corrected data from earlier gridded products in the areas of Petermann Glacier, HagenBræ and Sermilik Fjord assuming that sub-ice and fjord bathymetries roughly follow plausible Last Glacial Maximum ice flow patterns." Even if this certainly is an improvement likely closer to the truth, it is another assumption not based on real bathymetric data. For this reason, I do not think it is appropriate referring to it as a "correction". Instead "modified" is more appropriate.

- "You are totally right. Change applied."

Could this modification be shown graphically? It is not made fully clear how this assumption is technically making its way into the gridded compilation.

- "As examples we showed our modifications for Petermann and Ryder Glacier (Figure 4a-b) and Sermilik Fjord and Køge Bugt (Figure 4c-d). We explain the modification carried out in Section 2.3.3 (Data Modifications) and added more details on our proceedings to be more explicit."

On page 6, please clarify the last part of the sentence: ...."each other; combining surface elevation and ice thickness maps yields an ice bottom topography that even for grounded ice is not identical to the bedrock topography grid provided.". . .

- "... each other: combining surface elevation and ice thickness maps yields an ice bottom topography that is not identical to the bedrock topography grid provided for grounded ice areas."

On page 2 it is written: "This is particularly true for the northeast Greenland continental shelf, where a system of troughs provides a flux of warm water towards the floating ice tongues of Nioghalvfjerdsfjorden Glacier (also referred to as 79 North Glacier) and Zachariæ Isstrøm (Arndt et al. (2015), Wilson and Straneo (2015)). " Why is

this particularly true here, troughs of western Greenland seems equally pronounced and the same problem apply? We know less about the coupling between the fjords and the glacial troughs along the northwester sector of the Greenland coast, north of Upernavik), but this does not exclude them as being sensitive from influx of warmer subsurface water.

-"We agree and modified this passage, now referring to the northeast Greenland Shelf as "One of the key regions""

- "We did not include additional trough bathymetries yet because we are not aware of any other existing available bathymetric maps for the Greenland coastal regions yet. We are ready to include further data in revised/updated future versions of RTopo-2 though."

Please also note the supplement to this comment: http://www.earth-syst-sci-data-discuss.net/essd-2016-3/essd-2016-3-AC2-supplement.pdf

**Supplement:**

**A global, high-resolution data set of ice-sheet topography, cavity geometry, and ocean bathymetry**

Janin Schaffer[1], Ralph Timmermann[1], Jan Erik Arndt[1], Steen Savstrup Kristensen[2], Christoph Mayer[3], Mathieu Morlighem[4], and Daniel Steinhage[1]

[1]Alfred Wegener Institute, Helmholtz Centre for Polar and Marine Research, Bremerhaven, Germany
[2]DTU Technical University of Denmark, 2800 Lyngby, Denmark
[3]Bavarian Academy of Sciences and Humanities, Commission for Geodesy and Glaciology, Munich, Germany
[4]University of California, Irvine, Department of Earth System Science, Croul Hall, Irvine, California 92697-3100, USA

*Correspondence to:* J. Schaffer (Janin.Schaffer@awi.de) and R. Timmermann (Ralph.Timmermann@awi.de)

**Abstract.** The ocean plays an important role in modulating the mass balance of the polar ice sheets by interacting with the ice shelves in Antarctica and with the marine-terminating outlet glaciers in Greenland. Given that the flux of warm water onto the continental shelf and into the sub-ice cavities is steered by complex bathymetry, a detailed topography data set is an essential ingredient for models that address ice-ocean interaction. We followed the spirit of the global RTopo-1 data set and compiled consistent maps of global ocean bathymetry, upper and lower ice surface topographies and global surface height on a spherical grid with now 30-arc-seconds grid spacing . For this new data set, called RTopo-2, we used the General Bathymetric Chart of the Oceans (GEBCO_2014) as the backbone and added the International Bathymetric Chart of the Arctic Ocean version 3 (IBCAOv3) and the International Bathymetric Chart of the Southern Ocean (IBCSO) version 1. While RTopo-1 primarily aimed at a good and consistent representation of the Antarctic ice sheet, ice shelves and sub-ice cavities, RTopo-2 now also contains ice topographies of the Greenland ice sheet and outlet glaciers. In particular, we aimed at a good representation of the fjord and shelf bathymetry surrounding the Greenland continent. We modified  data from earlier gridded products in the areas of Petermann Glacier, Hagen Bræ and Sermilik Fjord assuming that sub-ice and fjord bathymetries roughly follow plausible Last Glacial Maximum ice flow patterns. For the continental shelf off northeast Greenland and the floating ice tongue of Nioghalvfjerdsfjorden Glacier at about 79°N, we incorporated a high-resolution digital bathymetry model considering original multibeam survey data for the region. Radar data for surface topographies of the floating ice tongues of Nioghalvfjerdsfjorden Glacier and Zachariæ Isstrøm have been obtained from the data centers of Technical University of Denmark (DTU), Operation Icebridge (NASA/NSF) and Alfred Wegener Institute (AWI). For the Antarctic ice sheet/ice shelves, RTopo-2 largely relies on the Bedmap-2 product but applies corrections for the geometry of Getz, Abbot and Fimbul ice shelf cavities. The data set is available in full and in regional subsets in NetCDF format from the PANGAEA database at doi:10.1594/PANGAEA.856844.

**1   Introduction**

Mass loss from the Greenland ice sheet presently accounts for about 10% of the observed global mean sea-level rise (Church et al., 2013). The ocean plays an important role in modulating the flow of ice by delivering heat to the marine-terminating outlet glaciers around Greenland (e.g., Seale et al., 2011; Straneo et al., 2012). The warming and accumulation of Atlantic Water in the
5   subpolar North Atlantic has been suggested to be the driver of the glaciers' retreat around the coast of Greenland (e.g., Straneo and Heimbach, 2013). The complex bathymetry in this region is thought to steer the flux of warm water of Atlantic origin from the open ocean onto the continental shelf towards the calving fronts of outlet glaciers and into the cavities below floating ice tongues. One of the key regions here is  the northeast Greenland continental shelf, where a system of troughs supports the  flux of warm water towards the floating ice tongues of Nioghalvfjerdsfjorden Glacier (also
10   referred to as 79°North Glacier) and Zachariæ Isstrøm (Arndt et al. (2015), Wilson and Straneo (2015)). Recently, these glaciers were observed to retreat and melt rapidly (Mouginot et al., 2015). In such regions detailed bathymetry data and consistent data sets of ice topographies are essential ingredients for studying the interaction between the ocean and the cryosphere.

Around Antarctica, research into ocean-cryosphere interaction has been an established field of science for several decades. Many aspects of water mass modification in the Southern Ocean's marginal seas can only be understood if the fluxes of heat
15   and freshwater at the base of the ice shelves surrounding the Antarctic continent are considered (e.g., Foldvik et al., 1985). Scientific interest has increased further with growing evidence that mass loss from the Antarctic Ice Sheet is accelerating (e.g., McMillan et al., 2014) and driven by enhanced ice shelf basal melting (Pritchard et al., 2012). There again, a well-constrained rendition of ocean bathymetry and cavity geometry is key to a successful analysis of field data and to a realistic representation of the relevant processes in numerical models.

20   The Refined Topography data set (RTopo-1; Timmermann et al., 2010) provides consistent maps of the global ocean bathymetry and the upper and lower ice surface topographies of the Antarctic ice sheet and shelves. Horizontal grid spacing  of these maps is 1-arc-minute. Based on RTopo-1, ocean general circulation models have successfully been used to simulate, e.g., Southern Ocean warming and increased ice shelf basal melting around Antarctica (Timmermann and Hellmer, 2013; Kusahara and Hasumi, 2013), the flow of Circumpolar Deep Water  onto the Amundsen Sea
25   continental shelf (Assmann et al., 2013; Nakayama et al., 2014a), and pathways of basal meltwater from Antarctic ice shelves (Kusahara and Hasumi, 2014; Nakayama et al., 2014b). Parts of RTopo-1 were used to compile improved maps of bedrock and ice topographies for Antarctica in Bedmap2 (Fretwell et al., 2013). The Greenland ice sheet, however, has remained a blank area in RTopo-1.

The aim of this paper is to present the newly compiled global topography data set RTopo-2, which provides a detailed
30   bathymetry for the continental shelf around Greenland and contains ice and bedrock surface topographies for Greenland and Antarctica as part of a global, self-consistent data set with a horizontal grid spacing  of 30-arc-seconds. In the following sections, we introduce the data used, the processing applied to each data set and the strategies followed for merging the data sets in a self-consistent way. We demonstrate the improvements achieved in RTopo-2 compared to previous products and discuss the most relevant caveats.

[Figure]

**Figure 1.** Sketch of a 2D vertical section along a floating glacier tongue/ice shelf with grounded and floating ice (white), sub-ice bedrock/ocean seafloor (brown), water in a subglacial cavity and the open ocean (blue). Lines indicate the bedrock topography (brown), the surface elevation (dark blue), and the ice base topography (black).

**2 Data sets and processing**

**2.1 Overview of RTopo-2 maps**

We followed the spirit of RTopo-1 and compiled global fields for

1. bedrock topography (ocean bathymetry; surface topography of continents; bedrock topography under grounded or floating ice)

2. surface elevation (upper ice surface height for Antarctic and Greenland ice sheets/ice shelves; bedrock elevation for ice-free continent; zero for ocean)

3. ice base topography for the Antarctic and Greenland ice sheets/ice shelves (ice draft for ice shelves and floating glaciers; zero in absence of ice)

4. a surface type mask that indicates open ocean, grounded ice (ice sheets), floating ice (ice shelves/floating glaciers), and bare land surface

5. positions of coastlines and ice shelf/floating glacier front lines.

The bedrock topography is identical to the surface elevation for ice-free land surface, and identical to the ice base topography for grounded ice (Fig. 1). Ice not connected to the Greenland or Antarctic ice sheet is not covered in our data set. Glaciers on subantarctic and Greenland islands are thus labeled as bare land surface with the surface elevation preserved. In contrast to RTopo-1, we now provide all maps with a horizontal grid spacing  of 30-arc-seconds.

[Figure]

**Figure 2.** Data sources for the ocean bathymetry in RTopo-2. Black areas denote transition zones between the data sets (source flag 20). Further explanations are given in Table 1.

**2.2  Data sources and merging procedure**

RTopo-2 has been compiled by combining various gridded data sets (Table 1) with different grid spacings , projections, and coverages (Fig. 2) into global maps. For data handling, interpolation and blending we developed a command script written in Interactive Data Language (IDL). Using a global bathymetry dataset (see below for details) as a backbone, regional grids have been created from various source data sets and subsequently merged into the existing fields. Interpolation from different projections to our geographic grid was based on Delaunay triangulation and subsequent linear interpolation. The regional "patches" have been blended into the existing fields using weight functions that - depending on the distance from the boundaries of the regional grids - vary between 0 and 1 and ensure a smooth transition between the two data sets without smoothing the topographies. As weight functions we used Hyperbolic Tangent functions with empirically derived length scales that have been cut off below values of 0.05 and above 0.95 to avoid overly long tails. This approach yields good results only when the two grids to be merged do not differ too strongly in the area of overlap, but with the data sets used here it was always possible to choose the location and width of the transition zone in a way that ensured a smooth blending.

For each of the newly incorporated regional grids, we had to ensure or enforce consistency with the existing topographies. The necessity for this step is quite obvious when independent data sets are combined; interpolation of discontinuous fields (an obvious example here is the discontinuity at ice shelf fronts) is another source for the creation of local inconsistencies that need to be cured. For RTopo-2, the term *consistency* implies that

– ice thickness $> 0$ in the ice-covered region (with ice thickness = surface height minus lower ice topography)

**Table 1.** Data sources for individual regions merged in RTopo-2. The index numbers correspond to the source flags in Fig. 2.

| | Region | Data obtained from |
|---|---|---|
| 1. | World Ocean bathymetry | GEBCO_2014 (Weatherall et al., 2015) |
| 2. | Southern Ocean bathymetry | IBCSO (Arndt et al., 2013) |
| 3. | Arctic Ocean bathymetry | IBCAOv3 (Jakobsson et al., 2012) |
| 4. | Antarctic ice sheet/shelves surface height and thickness and bedrock topography | Bedmap2 (Fretwell et al., 2013) |
| 5. | Greenland ice sheet/glaciers surface height and thickness and bedrock topography | Morlighem et al. (2014) (M-2014) |
| 6. | Fjord and shelf bathymetry close to the Greenland coast | Bamber et al. (2013) (B-2013) |
| 7. | Bathymetry on Northeast Greenland continental shelf | Arndt et al. (2015) (NEG_DBM) |
| 8. | Bathymetry in several narrow Greenland fjords and on parts of the Greenland continental shelf | artificial, see Merging strategy and Data corrections in Section 2.2.3 for details |
| 9. | Bathymetry for Getz and western Abbot Ice Shelf cavities | ALBMAP (Le Brocq et al., 2010) |
| 10. | Bathymetry for Fimbulisen cavity | Nøst (2004), Smedsrud et al. (2006) |
| 11. | Ice thickness for Nioghalvfjerdsfjorden Glacier and Zachariæ Isstrøm | DTU (Seroussi et al., 2011) Operation Icebridge (Allen et al., 2010, updated 2015) Alfred Wegener Institute (AWI) Mayer et al. (2000) |
| 12. | Contour of iceberg A-23A in Weddell Sea | Paul et al. (2015) |

- water column thickness > 0 in the open ocean and sub-ice cavities (with water column thickness = lower ice topography minus bedrock topography)

- water column thickness is zero, i.e. lower ice and bedrock topographies are identical, for grounded ice

- lower ice topography is negative (below sea level) and surface height is positive (above sea level) for ice shelves

- ice draft and thickness are zero outside ice-covered regions

- bedrock topography is below sea level in the ocean

- there are no enclosed gaps ('holes') in the ice sheet / ice shelves other than those associated with rock outcrops

- there are no water areas south of the coastline of Antarctica

These points may all seem trivial, but they are in fact not universally ensured in the gridded datasets available to date. Note that the surface type mask plays a key role in our algorithm; instead of being a merely diagnostic property, the surface type determines the conditions to which consistency is enforced. Choices that needed to be made include deciding which of the topographies should be trusted more, e.g., is bedrock from one source or lower ice topography from another source more

5   reliable, etc. These decisions were not always straightforward and are somewhat subjective; we give some of the reasoning in the sections discussing specific regional data sets below. In general, consistency and continuity have been valued higher than an exact rendition of the source data sets in RTopo-2.

**2.3   Bedrock and bathymetry data sets**

**2.3.1   World Ocean bathymetry**

[revised manuscript text omitted]

15   500 m, which is more likely to under- than to overestimate the true depth. Subsequently we smoothed over the artificial depth values by applying a moving average with a smoothing radius of 4 km taking into account the surrounding data points and the surface type mask.

Further to the east, the B-2013 representation of the continental shelf area in front of Ryder Glacier features a very steep gradient and a deep trough close to the coast, which appears unrealistic. We replaced some of the interpolated deep and adjacent

20   shallow parts with a smooth deep fjord/shelf bathymetry (Figs. 4a and 4b). In practice, we prescribed small patches with depths values of 750 m, 800 m and 900 m. Afterwards we smoothed the bedrock elevation within these areas by applying a moving average with a smoothing radius of 4 km. 
[revised manuscript text omitted]